# Hemodynamics Prior to Valve Replacement for Severe Aortic Stenosis and Pulmonary Hypertension during Long-Term Follow-Up

**DOI:** 10.3390/jcm10173878

**Published:** 2021-08-28

**Authors:** Lukas Weber, Hans Rickli, Philipp K. Haager, Lucas Joerg, Daniel Weilenmann, Joannis Chronis, Johannes Rigger, Marc Buser, Niklas F. Ehl, Micha T. Maeder

**Affiliations:** Cardiology Department, Kantonsspital St. Gallen, CH-9007 St. Gallen, Switzerland; lukas.weber@kssg.ch (L.W.); hans.rickli@kssg.ch (H.R.); philipp.haager@kssg.ch (P.K.H.); lucas.joerg@kssg.ch (L.J.); daniel.weilenmann@kssg.ch (D.W.); joannis.chronis@kssg.ch (J.C.); johannes.rigger@kssg.ch (J.R.); marc.buser@kssg.ch (M.B.); niklas.ehl@kssg.ch (N.F.E.)

**Keywords:** pulmonary hypertension, aortic valve replacement, right heart catheterization, persistent, pulmonary vascular resistance, pulmonary capacitance

## Abstract

(1) Background: Pulmonary hypertension after aortic valve replacement (AVR; post-AVR PH) carries a poor prognosis. We assessed the pre-AVR hemodynamic characteristics of patients with versus without post-AVR PH. (2) Methods: We studied 205 patients (mean age 75 ± 10 years) with severe AS (indexed aortic valve area 0.42 ± 0.12 cm^2^/m^2^, left ventricular ejection fraction 58 ± 11%) undergoing right heart catheterization (RHC) prior to surgical (70%) or transcatheter (30%) AVR. Echocardiography to assess post-AVR PH, defined as estimated systolic pulmonary artery pressure > 45 mmHg, was performed after a median follow-up of 15 months. (3) Results: There were 83/205 (40%) patients with pre-AVR PH (defined as mean pulmonary artery pressure (mPAP) ≥ 25 mmHg by RHC), and 24/205 patients (12%) had post-AVR PH (by echocardiography). Among the patients with post-AVR PH, 21/24 (88%) had already had pre-AVR PH. Despite similar indexed aortic valve area, patients with post-AVR PH had higher mPAP, mean pulmonary artery wedge pressure (mPAWP) and pulmonary vascular resistance (PVR), and lower pulmonary artery capacitance (PAC) than patients without. (4) Conclusions: Patients presenting with PH roughly one year post-AVR already had worse hemodynamic profiles in the pre-AVR RHC compared to those without, being characterized by higher mPAP, mPAWP, and PVR, and lower PAC despite similar AS severity.

## 1. Introduction

In patients with severe aortic stenosis (AS), pulmonary hypertension (PH) is common and associated with increased mortality after surgical (SAVR) or transcatheter (TAVR) aortic valve replacement (AVR) [1]. In patients with severe AS, PH typically is the result of left atrial pressure elevation due to the combined effects of left ventricular pressure overload on left ventricular and left atrial function and the development of functional mitral regurgitation. If left atrial pressure elevation is sustained and substantial, secondary pulmonary vascular remodeling may occur [1]. Studies have shown an early reduction in pulmonary pressure after TAVR, but this is often incomplete [2,3,4,5]. In a contemporary cohort, significant PH (defined as systolic pulmonary artery pressure (sPAP) >45 mmHg as assessed by echocardiography) one month after TAVR was found in one quarter of patients, and this was a marker of poor prognosis [4]. The persistence of PH after AVR is explained by the fact that PH in AS in not only the result of the outflow tract obstruction per se, but represents the synergistic effects of the downstream maladaptive changes to the left ventricle, the left atrium, the mitral valve, and the pulmonary vasculature [6], which more recently has been referred to as “cardiac damage” [7,8]. Although TAVR has been shown to result in an immediate reduction in pulmonary pressure in some patients [5] and improvement in right ventricular to pulmonary artery coupling [9], reversal of “cardiac damage” after relief of outflow tract obstructive may require time [10] and the damage may sometimes even be irreversible [1]. Therefore, the full impact of AVR on PH may become obvious only after several months. However, data on the longer-term effects of AVR on PH and the hemodynamic predictors of PH persisting several months or years post-AVR are very limited. In particular, there are no studies analyzing late post-AVR PH in relation to invasive pre-AVR hemodynamics. In the present study, we assessed the prevalence of PH several months after SAVR or TAVR in a cohort of patients who underwent left and right heart catheterization prior to surgery or intervention, respectively. We hypothesized that post-AVR PH was restricted to patients with pre-AVR PH, and in particular to patients with a precapillary component to PH.

## 2. Materials and Methods

### 2.1. Study Population

The present study was a retrospective subgroup analysis of prospectively and systematically collected data of patients with severe AS undergoing cardiac catheterization prior to AVR in a single center between January 2011 and January 2016 (entire cohort: *n* = 503) [11]. Right heart catheterization is routinely performed in all patients evaluated for AVR at the time of coronary angiography at our center. For this analysis, we included all 205 patients for whom an echocardiogram was available approximately one year post-AVR where the sPAP was reported. Of the other 298 patients, 45 patients had died in the first months after AVR, and in 253 patients, an echocardiogram with a reliable sPAP measurement was not available. The study was approved by the local ethics committee. A waiver of consent was granted. We have previously reported on other aspects of the hemodynamics in this population [11,12,13,14,15,16,17].

### 2.2. Cardiac Catheterization

Procedures were generally (>95%) performed in the morning in a fasting state and after withholding loop diuretics and renin–angiotensin system inhibitors. Patients underwent coronary angiography using 5 or 6 French catheters via a femoral or radial artery and right heart catheterization using 6 French Swan Ganz catheters via a femoral or brachial vein. The midthoracic level was used as the zero reference point. Right atrial pressure, right ventricular pressure, pulmonary artery pressure (PAP), and pulmonary artery wedge pressure were measured. The wedge position was confirmed by fluoroscopy and waveform analysis. Confirmation of wedge position by blood aspiration and blood gas analysis was not performed. Measurements were obtained at end-expiration, the mean pulmonary artery wedge pressure (mPAWP) was calculated over the entire cardiac cycle, and v waves were included to determine mPAWP. This practice leads to higher values compared to the measurement of the end-diastolic pulmonary artery wedge pressure [18]. However, for estimation of the impact of the left heart contribution to pulmonary pressures and calculation of PVR, respectively, mPAWP is preferred [19,20]. In patients with atrial fibrillation, at least five cardiac cycles were used to assess PAP and pulmonary artery wedge pressure (sinus rhythm: usually three cycles). Cardiac output (CO) was assessed using the indirect Fick method based on blood gases, which were collected in duplicate from the arterial access and the pulmonary artery. After completion of right heart catheterization, the aortic valve was crossed with a stiff wire in a subset of patients, and the left ventricular end-diastolic pressure (LVEDP) was measured using a pigtail catheter. All pressure readings were double-checked by the operator by manual review of the pressure tracings before they were entered into the report. The transpulmonary gradient (TPG) was calculated as mPAP-mPAWP. Pulmonary vascular resistance (PVR, in Wood units (WU)) was calculated as TPG/CO, the diastolic pressure gradient (DPG) was calculated as diastolic PAP—mPAWP, and pulmonary arterial capacitance (PAC) was calculated as stroke volume/(systolic PAP—diastolic PAP), where stroke volume is CO/heart rate.

### 2.3. Definition of Pulmonary Hypertension

Pulmonary hypertension was defined according to both the 2015 European Society of Cardiology (ESC)/European Respiratory Society (ERS) guidelines [21] and the 2018 PH World Symposium proposal [22]. According to the 2015 ESC/ERS guidelines, any PH is defined as mPAP ≥ 25 mmHg [21]. It is further classified as precapillary PH (mPAWP ≤ 15 mmHg, no PVR criterion), isolated postcapillary PH (IpcPH; mPAWP *>* 15 mmHg, PVR ≤ 3 WU and/or DPG < 7 mmHg), or combined pre- and postcapillary PH (CpcPH; mPAWP *>* 15 mmHg, PVR *>* 3 WU and/or DPG ≥ 7 mmHg) [21]. Given the recent controversy about use of the DPG for the definition of CpcPH [23] and the conflicting data on its prognostic value [24], and the fact that the application of the original 2015 ESC/ERS definition results in unclassifiable patients (i.e., those with discordant PVR and DPG: PVR ≤ 3 WU but DPG ≥ 7 mmHg or PVR > 3 WU but DPG < 7 mmHg) [25], we decided to use the PVR criterion only. According to the 2018 definition, precapillary PH is defined as mPAP > 20 mmHg, mPAWP ≤ 15 mmHg, and PVR ≥ 3 WU; IpcPH as mPAP > 20 mmHg, mPAWP > 15 mmHg, and PVR < 3 WU; and CpcPH as mPAP > 20 mmHg, mPAWP > 15 mmHg, and PVR ≥ 3 WU [22,23]. Patients with mPAP > 20 mmHg, mPAWP ≤ 15 mmHg, and PVR < 3 WU fulfil criteria for neither precapillary PH nor postcapillary PH. This group of patients is not explicitly addressed in the 2018 definition papers [22,23], and it is not clearly stated either whether any patient with mPAP > 20 mmHg should be classified as having PH. Therefore, we assumed that these patients do not have PH as there was not clear evidence for pulmonary vascular disease and/or elevated left atrial pressure [15]. 

### 2.4. Echocardiography and Follow-Up

All patients received an echocardiogram prior to cardiac catheterization as a basis for the referral and a follow-up study approximately one year after AVR. Echocardiograms were performed by experienced cardiologists according to contemporary guidelines but not according to a specific study protocol. The data were retrospectively obtained from the reports. The sPAP was estimated based on measurement of the peak tricuspid regurgitation velocity and the modified Bernoulli equation. It is our practice to use only tricuspid regurgitation signals with clearly visible peak velocities for sPAP estimation. Post-AVR PH was defined as sPAP > 45 mmHg based on the study by Masri et al. [4], who showed that this a prognostically relevant measure in the post-AVR setting. Right ventricular to pulmonary artery coupling was described as the tricuspid annular plane systolic excursion (TAPSE)/sPAP ratio [26].

### 2.5. Statistical Analysis

Categorical data are presented as numbers and percentages, and continuous data are reported as mean ± standard deviation or median (interquartile range) as appropriate. Clinical characteristics and echocardiographic and hemodynamic data in patients with and without post-AVR PH were compared using chi-square tests, unpaired *t*-tests, or Mann–Whitney U test as appropriate. Receiver operator characteristic (ROC) curves were constructed to assess the area under the curve (AUC) for single invasive hemodynamic parameters and PH according to the 2015 and 2018 definition as well as noninvasive sPAP to predict post-AVR PH. A *p*-value < 0.05 was considered statistically significant. Analyses were performed using SPSS statistical package version 20.0 (SPSS Inc., Chicago, IL, USA).

## 3. Results

### 3.1. Study Population

We studied 205 patients with a mean age of 75 ± 10 years, of which 50% were males. The mean indexed aortic valve area was 0.42 ± 0.12 cm^2^/m^2^ and the mean left ventricular ejection fraction (LVEF) was 58 ± 11%. Additional clinical and echocardiographic characteristics are shown in Table 1 and Table 2. 

### 3.2. Baseline Cardiac Catheterization

In the pre-AVR cardiac catheterization, the mPAP, mPAWP, PVR, and PAC values in the entire population were 25 ± 10 mmHg, 16 ± 8 mmHg, 2.1 ± 13 WU, and 3.4 ± 2.0 mL/mmHg. According to the 2015 definition, there were 83 (40%) patients with PH: 9 (4%) had precapillary PH, 44 (21%) had IpcPH, and 30 (15%) had CpcPH. According to the 2018 definition, there were 84 (41%) with PH: three (1%) had precapillary PH, 51 (25%) had IpcPH, and 30 (15%) had CpcPH (Table 2).

### 3.3. Aortic Valve Replacement

All patients underwent SAVR (143/205; 70%) or TAVR (62/205; 30%). In patients undergoing SAVR, the following additional procedures were performed: coronary bypass grafting (*n* = 45), surgery of the ascending aorta (*n* = 13), mitral valve reconstruction (*n* = 8), tricuspid valve reconstruction (*n* = 1), and Maze procedure (*n* = 7). Patients did not experience major cardiovascular events until the time of the follow-up echocardiogram, except one patient who experienced a nondisabling stroke within the first 30 days after SAVR.

### 3.4. Post-AVR PH Status

Follow-up echocardiograms were performed at a median interval of 15 (interquartile range, 12–18) months after SAVR or TAVR (Table 3). The mean sPAP was 33 ± 10 mmHg, and 24 (12%) patients had an sPAP > 45 mmHg and thereby fulfilled criteria for post-AVR PH. 

### 3.5. Pre-AVR Clinical Characteristics of Patients with versus without Post-AVR PH

Patients with post-AVR PH were older, more likely to be female, and had lower hemoglobin, estimated glomerular filtration rate, and forced expiratory volume within the first second that those without at the time of the pre-AVR assessment. In addition, patients with post-AVR PH were more symptomatic, had higher B-type natriuretic peptide (BNP), were more likely to take loop diuretics and spironolactone before AVR, and were more likely to undergo TAVR than patients without (Table 1). The peak early mitral inflow velocity to the peak early mitral annular velocity (E/e’) ratio was higher in patients with post-AVR PH compared to those without. There were trends towards lower LVEF and TAPSE, larger indexed left atrial area, more severe mitral regurgitation, and worse TAPSE/sPAP ratio in patients with post-AVR PH compared to those without but these failed to reach statistical significance (Table 2). However, the severity of AS as described by the indexed aortic valve area and the extent of coronary artery disease did not differ between patients with and without post-AVR PH.

### 3.6. Pre-AVR Invasive Hemodynamics of Patients with versus without Post-AVR PH

There were substantial and statistically highly significant differences in the key pre-AVR invasive hemodynamic parameters between patients with and without post-AVR PH (Table 2, Figure 1): patients with post-AVR had higher mean right atrial pressure, right ventricular end-diastolic pressure, mPAP, mPAWP, and PVR, and lower PAC, cardiac index, and stroke volume index than those without. In contrast, there were no significant differences in LVEDP and DPG between groups. Independently of the PH definition applied, patients with post-AVR already were more likely to have PH in the pre-AVR invasive assessment, i.e., 2015 definition: 21/24 (88%) versus 62/181 (34%; *p* < 0.001); 2018 definition: 22/24 (92%) versus 62/181 (34%; *p* < 0.001). In particular, the prevalence of CpcPH was markedly higher in patients with post-AVR PH (nearly 50% irrespective of the definition used) compared to those without (Table 2). On the other hand, 62/83 (75%; 2015 definition) or 62/84 (74%; 2018 definition) patients with pre-AVR PH did not have evidence of significant PH at follow-up.

### 3.7. Clinical Status and Echocardiography at Follow-Up

At the time of the follow-up echocardiogram, patients with post-AVR PH were still more symptomatic than those without (Table 3). Apart from a higher sPAP (by definition) they had lower TAPSE, lower TAPSE/sPAP ratio, larger indexed left atrial area, higher E/e’, and more severe mitral regurgitation (Table 3). Notably, these differences were more pronounced than in the pre-AVR echocardiogram (Figure 2). In contrast, the mean gradient across the aortic valve prosthesis, left ventricular size, and LVEF did not differ between groups.

### 3.8. Prediction of Post-AVR PH

The AUC values for the prediction of post-AVR PH for invasively assessed mPAP, mPAWP, PVR, and PAC were 0.82, 0.78, 0.78, and 0.82 (Figure 3A). The AUC values for the prediction of post-AVR PH for the 2015 and the 2018 PH definitions were 0.77 and 0.82 (Figure 3B). The AUC for sPAP as estimated from pre-AVR echocardiography was 0.74. 

## 4. Discussion

The present study provides the first detailed analysis of invasive pre-AVR hemodynamics in patients with severe AS presenting with post-AVR PH roughly one year after SAVR or TAVR. We were able to show that in a population with a pre-AVR PH prevalence of 40% as assessed by right heart catheterization and an overall uncomplicated clinical course in the initial months post-AVR, 12% met the criteria for significant post-AVR PH using an established echocardiographic definition. Importantly, 88% or 92% (depending on the PH definition applied) of these patients already had PH at the invasive pre-AVR assessment, and approximately 50% had a relevant precapillary component of PH (i.e., CpcPH in most cases). These patients with post-AVR PH remained more symptomatic after median follow-up of 15 months, and echocardiography suggested progressive left ventricular diastolic and left atrial dysfunction despite successful AVR. Notably, “true” precapillary PH (i.e., without a postcapillary component) was very rare. On the one hand, this is as expected in a population with left heart disease. On the other hand, recognition of a precapillary form of PH unrelated to AS would be very important, because symptoms of patients with this constellation may not improve after AVR. However, our study clearly shows that also in AS patients with pre-AVR CpcPH, residual PH can persist post-AVR.

Recent invasive studies have reported a prevalence of PH between 48% and 75% in patients with severe AS evaluated prior to AVR [3,5,11,27]. The prevalence of baseline PH in the present cohort was therefore relatively low. However, this is explained by two factors: First, we studied a mixed SAVR and TAVR population [11] rather than a pure, typically old and comorbid TAVR population as in other studies [3,5,27]. Second, the present study population already represented a positively selected group from our original cohort. In our entire SAVR and TAVR cohort, the PH prevalence was 48% [11]. The sickest patients died during the early months after AVR, and these patients were not available for a follow-up. Still, there were patients with an unfavorable evolution. Thus, our study may even underestimate the prevalence and extent of post-AVR PH.

We showed that approximately 75% of patients with pre-AVR PH presented without significant PH by echocardiography roughly one year after AVR. This underscores that AVR is an effective treatment not only for the improvement of symptoms and prognosis [28] but also for reducing pulmonary pressure. Schewel et al. [5] reported that 26% of patients with precapillary PH, 29% of patients with IpcPH and 48% of patients with CpcPH experienced a reduction in invasively assessed mPAP immediately following TAVR. O’ Sullivan et al. [3] observed a reduction in sPAP (by echocardiography) from pre-AVR to discharge in patients with invasively diagnosed IpcPH (from 50 to 45 mmHg) and CpcPH (from 58 to 50 mmHg), whereas such a reduction was not observed in those with precapillary PH (from 49 to 50 mmHg). Masri et al. [4] reported that in 63% of patients with pre-AVR PH (invasively assessed mPAP ≥ 25 mmHg), there was a reduction from an invasively assessed sPAP of 55 mmHg (and a noninvasively assessed sPAP of 42 mmHg) to a noninvasively assessed sPAP ≤ 45 mmHg within one month. On the other hand, in 37% of patients with pre-AVR PH, an invasively assessed sPAP of 64 mmHg (and a noninvasively assessed sPAP of 59 mmHg) did not decrease below the noninvasive sPAP cut-off of 45 mmHg within a month. Thus, all these studies looked at the immediate or short-term effects of AVR. This is likely too short an observation window to observe the full effect of AVR on left ventricular, left atrial, and pulmonary vascular remodeling [10]. There are studies reporting longer-term post-AVR sPAP data, but in these studies, there was no invasive pre-AVR assessment [2,29,30]. For example, Sinning et al. [2] found a decrease in sPAP from 66 mmHg to 60 mmHg at three months after TAVR in patients with severe pre-TAVR PH (defined as sPAP > 60 mmHg by echocardiography). A recent study found a reduction in the prevalence of PH (defined as a peak tricuspid regurgitation velocity ≥2.8 m/s, corresponding to an sPAP of approximately 31 mmHg) from 51% to 29% within 6 months after TAVR (*n* = 170) [30]. Another study reported a PH prevalence (defined as sPAP ≥ 50 mmHg) of 40% among 122 patients undergoing TAVR [29]. Six months post-TAVR, 57% of these patients experienced a reduction in sPAP to below 50 mmHg [29].

The present study with an invasive pre-AVR assessment and a median interval of 15 months between AVR and echocardiography therefore significantly adds to the existing literature. Whereas 75% of patients with pre-AVR PH and 88% of the entire population did not have relevant PH at follow-up, 25% of patients with pre-AVR PH and 12% of the entire population presented with at least moderate post-AVR PH. Obviously, AVR was not able to induce a full reversal of “cardiac damage” in these patients. We did not perform right heart catheterization at follow-up, which would have been very instructive but would have been hard to justify because treatment options for group 2 PH are very limited [23]. Our echocardiographic data do suggest, however, progressive left ventricular and left atrial dysfunction with functional mitral regurgitation. In this situation, mPAWP will remain elevated, which in turn may prevent reverse pulmonary vascular remodeling. Little is known about the characteristics of the pulmonary vasculature in CpcPH. An autopsy study in patients with PH in the context of heart failure and a broad LVEF spectrum revealed global pulmonary vascular remodeling, in particular remodeling of the veins, which is in contrast to the changes observed in patients with pulmonary arterial hypertension, which are confined to the precapillary vessels [31]. The substantial reductions in PVR in patients with advanced heart failure and CpcPH treated with left ventricular assist devices suggest that reverse pulmonary vascular remodeling following efficient reduction in mPAWP is possible [32]. The exact mechanisms of PH reversal in patients with AS are unknown, however. Interestingly, pre-AVR LVEDP was similar in patients with and without post-AVR PH, which may indicate a similar degree of diastolic dysfunction in both groups, although this did not fully correlate with the E/e’. However, mPAWP and PVR were significantly higher in patients later presenting with post-AVR PH, suggesting that these patients had more advanced left atrial dysfunction, more severe mitral regurgitation, and more advanced pulmonary vascular remodeling. At follow-up, there was a further increase in E/e’ and left atrial size, which may be interpreted as progressive left ventricular diastolic and left atrial dysfunction. This may be explained by unchanged or progressive left ventricular fibrosis, which in advanced stages of AS can be irreversible [33]. Importantly, pre-AVR invasive hemodynamics offered reasonable accuracy in predicting post-AVR PH. In particular, mPAP (which is hard to estimate noninvasively in individual patients) and PAC, a measure integrating mPAWP and PVR [34,35], had AUC values of more than 0.80 to predict post-AVR PH.

Although the present study was relatively small and probably excluded the most severe cases of post-AVR PH, it provides insights into potential mechanism of this condition, which is very hard to treat [1]. In a recent randomized trial, sildenafil failed to provide a benefit in patients with PH (IpcPH and CpcPH) that persisted despite correction of valve disease [36]. The concept of “cardiac damage” in AS highlights the fact that patients with pre-AVR PH have a higher long-term risk of death than AS patients without PH [7]. The phenomenon of post-AVR PH shows that in at least some patients, AVR is not able to fully reverse “cardiac damage”. At the moment, it is unknown which type and extent of “cardiac damage” is so advanced that the risk of AVR outweighs the benefits. Whether a clinical trial addressing this issue in a randomized manner will ever be performed is questionable. Most data on the “cardiac damage” concept are derived from noninvasive studies [8]. Therefore, detailed phenotyping of AVR candidates including a detailed invasive hemodynamic assessment may help in decision making and understanding the post-AVR course. Current guidelines propose that in “asymptomatic” patients with severe AS and an sPAP > 60 mmHg AVR should be considered (class IIa indication) [37]. Such patients may have already advanced pulmonary vascular disease that may be not be fully reversible after AVR. Systematic pre-AVR right heart catheterization in large cohorts may help to identify the patients at risk for post-AVR PH. Notwithstanding, early recognition of AS and regular follow-up will be a prerequisite. Research efforts will then have to concentrate on the optimal timing and selection of AS patients for “early” AVR and adjunct therapies attenuating unfavourable AS-related cardiac remodeling [1].

### Limitations

This study has several limitations. First, the number of patients was relatively low. Still, it is an invasive study, and due to the relatively long follow-up it provides new insights into the natural history of PH post-AVR. Second, echocardiograms were performed in clinical routine and not according to a research protocol. We had access to the relevant data, but not all parameters of interest were measured in a systematic manner, which led to a selection of patients. Third, data on cardiac catheterization were obtained from reports rather than directly from pressure tracings. However, our center has a long tradition of measuring invasive hemodynamics in patients with valve disease, and all cardiologists performing these examinations were experienced in this regard. Fourth, we used the indirect Fick method to assess cardiac output, which is subject to error since oxygen consumption is often inaccurately estimated [38]. This can affect all CO-based measures, including PAC and PVR. However, the indirect Fick method is routinely used in clinical practice. Fifth, the definition of post-AVR PH relied on echocardiography only. It is well known that estimation of sPAP by the peak tricuspid regurgitation velocity is only moderately accurate to predict sPAP and mPAP in individual patients. In addition, echocardiography has a limited ability to differentiate precapillary from postcapillary PH. Still, we used an established definition of post-AVR PH, which has previously been shown to predict prognosis [4]. Finally, the treatment modality (SAVR versus TAVR) clearly differed between those with and without post-AVR PH. This was most likely driven by the pre-AVR clinical and hemodynamic condition, but an effect of the mode of AVR on pulmonary pressure trajectories cannot be excluded, although this seems unlikely.

## 5. Conclusions

Patients undergoing AVR for severe AS and presenting with PH roughly one year post-AVR already had a significantly worse hemodynamic profiles in the pre-AVR RHC compared to patients without post-AVR PH, being characterized by higher mPAP, mPAWP, and PVR, and lower PAC despite similar AS severity. This underscores the prognostic importance of the pre-AVR RHC.

## Figures and Tables

**Figure 1 jcm-10-03878-f001:**
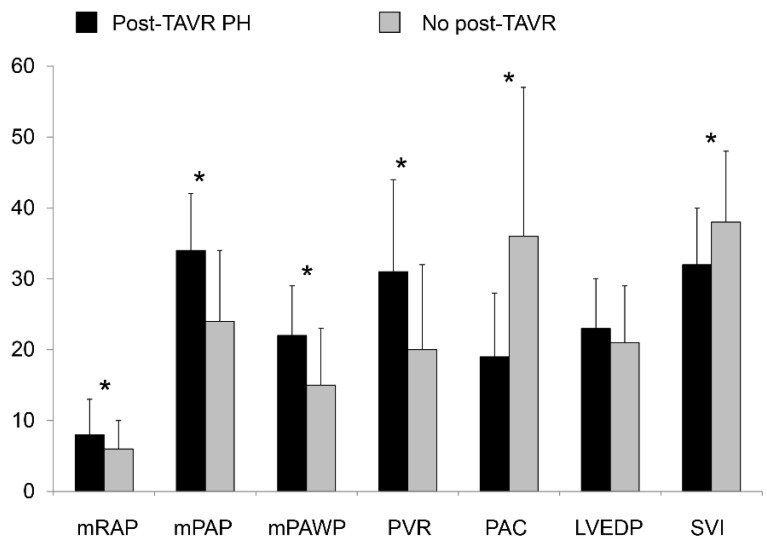
Hemodynamics prior to aortic valve replacement in patients with and without pulmonary hypertension several months after aortic valve replacement (post-AVR PH, defined as systolic pulmonary artery pressure > 45 mmHg by echocardiography). Bar graphs represent means ± standard deviations. * *p* < 0.01.LVEDP = left ventricular end-diastolic pressure; mPAP = mean pulmonary artery pressure; mPAWP = mean pulmonary artery wedge pressure; mRAP = mean right atrial pressure; PAC = pulmonary artery capacitance; PVR = pulmonary vascular resistance; SVI = stroke volume index. The scale is mmHg, Wood units × 10, mL/mmHg × 10, and mL/m^2^.

**Figure 2 jcm-10-03878-f002:**
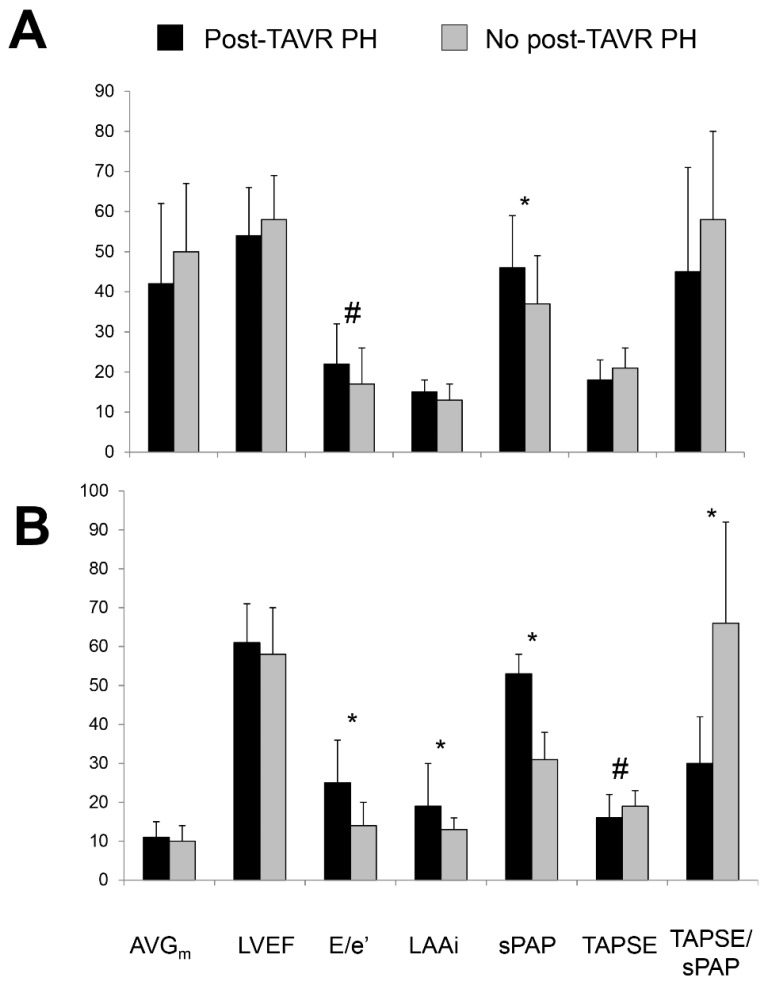
Key echocardiographic parameters prior to aortic valve replacement (panel (**A**)) and several months after aortic valve replacement (panel (**B**)) in patients with and without pulmonary hypertension several months after aortic valve replacement (post-AVR PH, defined as systolic pulmonary artery pressure > 45 mmHg by echocardiography). Bar graphs represent means ± standard deviations. * *p* < 0.01, # *p* < 0.05. AVG_m_ = mean aortic valve gradient, E/e’ = ratio of peak early mitral inflow velocity to peak early mitral annular velocity, LAAi = indexed left atrial area, LVEF = left ventricular ejection fraction, sPAP = systolic pulmonary artery pressure, TAPSE = tricuspid annular plane systolic excursion, TPASE/sPAP = ratio of TAPSE to sPAP as a noninvasive measure of right ventricular to pulmonary artery coupling. The scale is mmHg, %, dimensionless, cm^2^/m^2^, mmHg, mm, mm/mmHg × 100.

**Figure 3 jcm-10-03878-f003:**
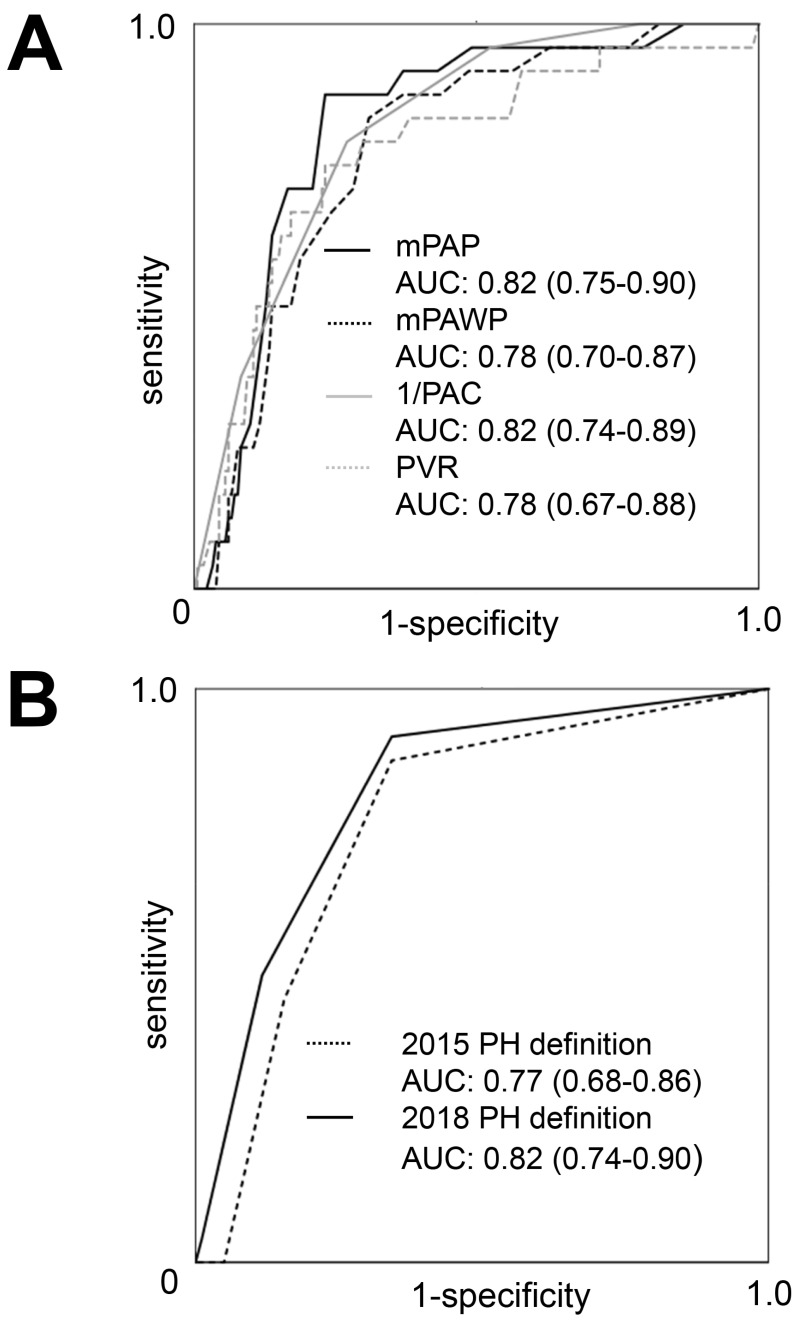
Receiver operator characteristics curves showing areas under the curve (AUC; with 95% confidence intervals) for the prediction of post-AVR PH for mean pulmonary artery pressure (mPAP), mean pulmonary artery wedge pressure (mPAWP), pulmonary artery capacitance (PAC), and pulmonary vascular resistance (PVR; all in panel (**A**)) and pulmonary hypertension (PH) according to the 2015 and 2018 definitions (panel (**B**)). Post-AVR PH = pulmonary hypertension several months after aortic valve replacement (defined as systolic pulmonary artery pressure > 45 mmHg by echocardiography).

**Table 1 jcm-10-03878-t001:** Pre-aortic valve replacement clinical characteristics of the entire study population and patients with versus without pulmonary hypertension after aortic valve replacement (post-AVR PH).

	All Patients (*n* = 205)	Post-AVR PH (*n* = 24)	No Post-AVR PH (*n* = 181)	*p* Value
Age (years)	75 ± 10	81 ± 6	74 ± 10	0.001
Gender (male)	102 (50%)	6 (25%)	96 (53%)	0.01
Body mass index (kg/m^2^)	27.3 ± 4.8	27.6 ± 4.5	27.2 ± 4.9	0.69
Body surface area (m^2^)	1.84 ± 0.22	1.78 ± 0.20	1.85 ± 0.23	0.14
eGFR (mL/min/1.73 m^2^)	72 ± 27	60 ± 19	73 ± 28	0.03
Hemoglobin (g/L)	134 ± 17	127 ± 16	135 ± 17	0.03
Diabetes	29 (14%)	6 (25%)	23 (13%)	0.10
Stroke	10 (5%)	1 (4%)	9 (5%)	0.86
Chronic obstructive lung disease	18 (9%)	2 (8%)	16 (9%)	0.93
FEV1 (% predicted)	89 ± 21	80 ± 19	90 ± 21	0.03
**Heart rhythm**				0.58
Sinus rhythm	178 (87%)	19 (79%)	159 (88%)	
Atrial fibrillation	19 (9%)	4 (17%)	15 (8%)	
pacemaker	8 (4%)	1 (4%)	7 (4%)	
Heart rate (bpm)	70 ± 13	73 ± 16	69 ± 12	0.14
**Medication**				
Oral anticoagulation	35 (17%)	9 (38%)	26 (14%)	0.05
Aspirin	112 (55%)	14 (58%)	98 (54%)	0.70
Loop diuretics	92 (45%)	19 (79%)	73 (40%)	<0.001
Betablocker	88 (43%)	13 (54%)	75 (41%)	0.24
ACEI/ARB	99 (48%)	10 (42%)	89 (49%)	0.49
Digoxin	17 (8%)	4 (17%)	13 (7%)	0.11
Spironolactone	10 (5%)	4 (17%)	6 (3%)	0.004
**Symptoms**				
Dyspnea NYHA class				0.008
I	36 (18%)	3 (12%)	33 (18%)	
II	109 (53%)	9 (38%)	100 (55%)	
III	51 (25%)	8 (33%)	43 (24%)	
IV	9 (4%)	4 (17%)	5 (3%)	
**B-type natriuretic peptide (ng/L)**	180 (75–448)	445 (256–779)	146 (59–333)	<0.001
**Mode of AVR**				<0.001
Surgical AVR	143 (70%)	7 (29%)	136 (75%)	
Transcatheter AVR	62 (30%)	17 (71%)	45 (25%)	

Data are given as numbers and percentages, mean ± standard deviation, or median (interquartile range). ACEI/ARB = angiotensin converting enzyme inhibitor/angiotensin receptor blocker; AVR = aortic valve replacement; eGFR = estimated glomerular filtration rate: FEV1 = forced expiratory volume within the first second; NYHA = New York Heart Association; PH: pulmonary hypertension.

**Table 2 jcm-10-03878-t002:** Pre-aortic valve replacement data from echocardiography and cardiac catheterization of the entire study population and patients with versus without pulmonary hypertension after aortic valve replacement (post-AVR PH).

	All Patients (*n* = 205)	Post-AVR PH (*n* = 24)	No Post-AVR PH (*n* = 181)	*p* Value
**Echocardiography**				
Left ventricular end-diastolic diameter (mm)	46 ± 8	44 ± 8	47 ± 8	0.18
Indexed left ventricular end-diastolic diameter (mm/m^2^)	26 ± 4	25 ± 4	26 ± 5	0.68
Left ventricular ejection fraction (%)	58 ± 11	54 ± 12	58 ± 11	0.05
E/e’	17.3 ± 9.1	21.7 ± 9.7	16.5 ± 8.9	0.04
Left atrial area (cm^2^)	25 ± 6	26 ± 6	24 ± 6	0.29
Indexed left atrial area (cm^2^/m^2^)	13 ± 3	15 ± 3	13 ± 4	0.08
TAPSE (mm)	21 ± 5	18 ± 5	21 ± 5	0.08
Estimated sPAP (mmHg)	39 ± 12	46 ± 13	37 ± 12	0.006
TAPSE/sPAP (mm/mmHg)	0.56 ± 0.23	0.45 ± 0.26	0.58 ± 0.22	0.08
Mean aortic valve gradient (mmHg)	49 ± 18	42 ± 20	50 ± 17	0.03
Aortic valve area (cm^2^)	0.77 ± 0.22	0.76 ± 0.19	0.77 ± 0.23	0.75
Indexed aortic valve area (cm^2^/m^2^)	0.42 ± 0.12	0.43 ± 0.11	0.42 ± 0.12	0.69
Aortic regurgitation (at least moderate)	24 (12%)	1 (4%)	23 (13%)	0.21
Mitral regurgitation				0.08
no	100 (49%)	6 (25%)	94 (52%)	
mild	87 (42%)	15 (63%)	72 (39%)	
moderate	16 (8%)	3 (12%)	13 (8%)	
severe	2 (1%)	0	2 (1%)	
**Coronary artery disease **				1.0
No coronary artery disease	123 (60%)	11 (46%)	112 (62%)	
1-vessel disease	35 (17%)	3 (12%)	32 (18%)	
2-vessel disease	23 (11%)	6 (25%)	17 (9%)	
3-vessel disease	24 (12%)	4 (17%)	20 (11%)	
**Invasive hemodynamics**				
Mean right atrial pressure (mmHg)	6 ± 4	8 ± 5	6 ± 4	0.009
Right ventricular end-diastolic pressure (mmHg)	8 ± 4	10 ± 5	7 ± 4	0.004
sPAP (mmHg)	39 ± 15	52 ± 11	37 ± 14	<0.001
dPAP (mmHg)	15 ± 8	21 ± 7	14 ± 8	<0.001
mPAP (mmHg)	25 ± 10	34 ± 8	24 ± 10	<0.001
mPAWP (mmHg)	16 ± 8	22 ± 7	15 ± 8	<0.001
Transpulmonary gradient (mmHg)	9 ± 4	12 ± 4	9 ± 4	0.002
Pulmonary vascular resistance (Wood units)	2.1 ± 1.3	3.1 ± 1.3	2.0 ± 1.2	<0.001
Diastolic pressure gradient (mmHg)	−1 (−3–1)	−2 (−5–3)	0 (−3–1)	0.35
Pulmonary artery capacitance (mL/mmHg)	3.4 ± 2.0	1.9 ± 0.9	3.6 ± 2.1	<0.001
Left ventricular end-diastolic pressure (mmHg)	21 ± 8 (*n* = 146)	23 ± 7 (*n* = 17)	21 ± 8 (*n* = 129)	0.41
Systolic aortic pressure (mmHg)	148 ± 26	161 ± 28	146 ± 25	0.007
Diastolic aortic pressure (mmHg)	68 ± 12	71 ± 13	68 ± 11	0.21
Mean aortic pressure (mmHg)	100 ± 15	106 ± 16	99 ± 14	0.03
Systemic vascular resistance (Wood units)	21.0 ± 5.7	25.0 ± 5.6	20.5 ± 5.5	<0.001
Arterial oxygen saturation (%)	95 (94–97)	94 (92–96)	95 (94–97)	0.03
Mixed venous oxygen saturation (%)	69 (64–73)	64 (58–7)	70 (65–73)	<0.001
Cardiac output (L/min)	4.6 ± 1.0	3.9 ± 0.7	4.7 ± 1.0	<0.001
Cardiac index (L/min/m^2^)	2.5 ± 0.5	2.2 ± 0.4	2.6 ± 0.5	0.003
Stroke volume index (mL/m^2^)	37 ± 10	32 ± 8	38 ± 10	0.003
PH classification 2015				<0.001
no PH	122 (60%)	3 (12%)	119 (66%)	
isolated postcapillary PH	44 (21%)	10 (42%)	34 (19%)	
combined pre- and postcapillary PH	30 (15%)	11 (46%)	19 (10%)	
precapillary PH	9 (4%)	0	9 (5%)	
PH classification 2018				<0.001
no PH	121 (59%)	2 (8%)	119 (66%)	
isolated postcapillary PH	51 (25%)	10 (42%)	41 (23%)	
combined pre- and postcapillary PH	30 (15%)	11 (46%)	19 (10%)	
precapillary PH	3 (1%)	1 (4%)	2 (1%)	

Data are given as numbers and percentages, mean ± standard deviation, and/or median (interquartile range). dPAP = diastolic pulmonary artery pressure; E/e’ = ratio of peak early mitral inflow velocity to peak early mitral annular velocity; mPAP = mean pulmonary artery pressure; mPAWP = mean pulmonary artery wedge pressure; sPAP = systolic pulmonary artery pressure; TAPSE = tricuspid annular plane systolic excursion.

**Table 3 jcm-10-03878-t003:** Symptoms and echocardiography at follow-up in all patients and patients with versus without pulmonary hypertension after aortic valve replacement (post-AVR PH).

	All Patients (*n* = 205)	Post-AVR PH (*n* = 24)	No Post-AVR PH (*n* = 181)	*p* Value
**Symptoms**				
Dyspnea NYHA class				<0.001
I	130 (64%)	11 (46%)	119 (66%)	
II	60 (29%)	7 (29%)	53 (29%)	
III	13 (6%)	4 (17%)	9 (5%)	
IV	2 (1%)	2 (8%)	0	
**Echocardiography**				
Left ventricular end-diastolic diameter (mm)	47 ± 7	47 ± 8	46 ± 7	0.68
Indexed left ventricular end-diastolic diameter (mm/m^2^)	26 ± 4	27 ± 5	25 ± 4	0.20
Left atrial area (cm^2^)	25 ± 9	36 ± 20	24 ± 7	<0.001
Indexed left atrial area (cm^2^/m^2^)	14 ± 5	19 ± 11	13 ± 3	<0.001
E/e’	15.5 ± 7.8	25.2 ± 10.6	14.0 ± 6.1	<0.001
Left ventricular ejection fraction (%)	61 ± 10	58 ± 12	61 ± 9	0.15
Mean aortic valve gradient (mmHg)	11 ± 4	10 ± 4	11 ± 4	0.13
Mitral regurgitation				0.002
no	98 (48%)	6 (25%)	92 (51%)	
mild	93 (45%)	13 (54%)	80 (44%)	
moderate	13 (6%)	4 (17%)	9 (5%)	
severe	1 (1%)	1 (4%)	0	
sPAP (mmHg)	33 ± 10	53 ± 5	31 ± 7	<0.001
TAPSE (mm)	18 ± 4	16 ± 6	19 ± 4	0.03
TAPSE/sPAP (mm/mmHg)	0.61 ± 0.27	0.30 ± 0.12	0.66 ± 0.26	<0.001

Data are given as number and percentages or mean ± standard deviation. E/e’ = ratio of peak early mitral inflow velocity to peak early mitral annular velocity; NYHA = New York _Heart Association; sPAP = systolic pulmonary artery pressure; TAPSE = tricuspid annular plane systolic excursion.

## Data Availability

The data underlying this article are available in the article.

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
