# Peer review of "Hemodynamics Prior to Valve Replacement for Severe Aortic Stenosis and Pulmonary Hypertension during Long-Term Follow-Up"

_jcm, 2021, doi:10.3390/jcm10173878_

Round 1

Reviewer 1 Report

In the manuscript entitled: “Hemodynamics prior to Valve Replacement for Severe Aortic Stenosis and Pulmonary Hypertension during Long-term Follow-up ” Weber et al. preformed a retrospective analysis using data from a single center. The authors described and compared patient population with and without pulmonary hypertension over a year after aortic valve replacement and assessed pre-procedure invasive hemodynamic to identified associated predictors. The analysis included 205 patients who underwent either SAVR or TAVR. The authors concluded that patients with pulmonary hypertension about one year after AVR had worse hemodynamic profile before the procedure, characterized by higher mPAP, mPAWP and PVR.

First, the authors should be praised for their professional and comprehensive work, my congrats.

Second, few minor comments:

  1. Pre AVR patient’s characteristic are presented in detail. However, it is not clear enough that Table 2 is a baseline echocardiographic and invasive hemodynamic data. Please consider changing the tables’ headline.
  2. Invasive hemodynamic provide detailed and mechanistic data (Pre-capillary, post-capillary and combined). One of the interesting findings in my opinion is that among 88% of patients who had pulmonary hypertension pre-AVR, only 4% (or none, depending on classification) had pre-capillary pulmonary hypertension. I feel that this point can be further discussed.
  3. The criteria are different for invasive and non-invasive pulmonary hypertension as the author discussed in detail. I think that a comparison between the two methods to evaluate baseline (pre AVR) pulmonary hypertension is important, especially due to the fact that only non-invasive measurements were used to compare with non-invasive follow-up measurements.
  4. This study included both trans-catheter and surgical interventional approaches for AVR. 71% of post-AVR pulmonary hypertension were treated by TAVR. Is it related to the procedure? Is it related to old age and co-morbidities? I would further discuss this issue, or at least highlight it in the limitation section.

Best of luck

Reviewer 2 Report

The authors showed that patients who develop post-AVR pulmonary hypertension often have pre-AVR pulmonary hypertension caused by pulmonary vascular remodeling.

There is no objection to this.

What is the reason for preoperative pulmonary hypertension? Age, gender, renal function, hypertension, atrial fibrillation, low output, disease period...? Please consider something from your own data or literature.
